# *Satureja montana* L. Essential Oils: Chemical Profiles/Phytochemical Screening, Antimicrobial Activity and O/W NanoEmulsion Formulations

**DOI:** 10.3390/pharmaceutics12010007

**Published:** 2019-12-19

**Authors:** Alessandro Maccelli, Luca Vitanza, Anna Imbriano, Caterina Fraschetti, Antonello Filippi, Paola Goldoni, Linda Maurizi, Maria Grazia Ammendolia, Maria Elisa Crestoni, Simonetta Fornarini, Luigi Menghini, Maria Carafa, Carlotta Marianecci, Catia Longhi, Federica Rinaldi

**Affiliations:** 1Dipartimento di Chimica e Tecnologie del Farmaco, Sapienza Università di Roma-Piazzale Aldo Moro 5, 00185 Roma, Italy; alessandro.maccelli@uniroma1.it (A.M.); anna.imbriano@uniroma1.it (A.I.); caterina.fraschetti@uniroma1.it (C.F.); antonello.filippi@uniroma1.it (A.F.); mariaelisa.crestoni@uniroma1.it (M.E.C.); simonetta.fornarini@uniroma1.it (S.F.); maria.carafa@uniroma1.it (M.C.); carlotta.marianecci@uniroma1.it (C.M.); federica.rinaldi@uniroma1.it (F.R.); 2Dipartimento di Sanità Pubblica e Malattie Infettive, Sapienza Università di Roma, Piazzale Aldo Moro 5, 00185 Roma, Italy; luca.vitanza@uniroma1.it (L.V.); paola.goldoni@uniroma1.it (P.G.); lindamaurizi92@gmail.com (L.M.); 3National Center of Innovative Technologies in Public Health, Italian National Institute of Health, Viale Regina Elena 299, 00161 Rome, Italy; maria.ammendolia@iss.it; 4Dipartimento di Farmacia, Università G. d’Annunzio Chieti-Pescara, Via dei Vestini 31, 66100 Chieti, Italy; luigi.menghini@unich.it; 5Center for Life Nano Science@Sapienza, Fondazione Istituto Italiano di Tecnologia, Viale Regina Elena 291, 00161 Rome, Italy

**Keywords:** *Satureja montana* L., essential oils, nanoformulation, nanoemulsions, antibacterial activity, FT-ICR mass spectrometry

## Abstract

Chemical fingerprints of four different *Satureja montana* L. essential oils (SEOs) were assayed by an untargeted metabolomics approach based on Fourier-transform ion cyclotron resonance (FT-ICR) mass spectrometry (MS) coupled with either electrospray ionization or atmospheric pressure chemical ionization ion sources. Analysis and relative quantification of the non-polar volatile fraction were conducted by gas chromatography (GC) coupled to MS. FT-ICR MS confirmed significant differences in the polar metabolite composition, while GC-MS analyses confirmed slight fluctuations in the relative amount of major terpenes and terpenoids, known to play a key role in antimicrobial mechanisms. Oil in eater (O/W) nanoemulsions (NEs) composed by SEOs and Tween 20 or Tween 80 were prepared and analyzed in terms of hydrodynamic diameter, ζ-potential and polydispersity index. The results confirm the formation of stable NEs homogeneous in size. Minimum inhibitory and minimum bactericidal concentrations of SEOs were determined towards Gram-positive (*Listeria monocytogenes*, *Staphylococcus aureus*, *Staphylococcus haemolyticus*) and Gram-negative clinical isolates (*Escherichia coli*, *Klebsiella pneumoniae*, *Pseudomonas aeruginosa,* and *Serratia marcescens*). Commercial SEO showed strongest antibacterial activity, while SEO 3 was found to be the most active among the lab made extractions. MIC and MBC values ranged from 0.39 to 6.25 mg·mL^−1^. Furthermore, a SEO structured in NEs formulation was able to preserve and improve antimicrobial activity.

## 1. Introduction

Medicinal plants have been used since ancient times as one of the main sources of therapeutic agents. Of the approximately 300,000 terrestrial plants that have been classified taxonomically, more than 10,000 plants are used for medical purposes [1]. A large number of research papers (exceeding 3000) and reviews (>300) published just in early 2019 (data retrieved from Scopus, July 2019; search: Medicinal plants) prove the great scientific interest attached to this area. On the other hand, many natural active compounds show low hydrophilicity and intrinsic dissolution rate, low absorption, poor pharmacokinetics or physical/chemical instability. These features limit their clinical use due to the need of high doses and/or repeated administrations. In the case of essential oils (EOs), the complex chemical composition together with high volatility and instability are further limitations.

In order to obtain better biopharmaceutical properties and to reduce doses and side effects, a great effort is needed to develop suitable delivery systems. A recent review reports some application of extract and EO nanoformulations with remarkable advantages over conventional ones, such as increased solubility and stability, higher permeation rate, better bioavailability, and sustained release [2]. The genus *Satureja* contains about 200 species of aromatic and medicinal herbs [3], which spontaneously grow in the Middle East and Mediterranean European regions, West Asia, North Africa, and South America [4]. Particularly, *Satureja montana* (*f.* Lamiaceae, *sf.* Nepetoideae, *t.* Mentheae), commonly known as winter or mountain savory, is a perennial small herb native to warm arid, sunny, and rocky regions in the Mediterranean area of Southern Europe and in Northern Africa [5,6]. In the Lamiaceae species, that can be considered as one of the richest in medicinal plant [7], *Satureja* essential oil (SEO) provides the basis for a wide range of biological and industrial applications with antiseptic, antioxidant, antifungal, carminative, and digestive properties [8,9], thanks to the content of biologically active phytochemicals. Furthermore, it is often used in Mediterranean cooking preparations and even as natural antibacterial agent in food packaging [10,11]. Recently, natural substances such as essential oils have attracted the attention of researchers due to their great beneficial effects [12,13]. Moreover, it has been estimated that the SEO yield from the fresh plant can exceed 5% [3], revealing a high biological potential. The volatile fraction is mainly characterized by oxygenated monoterpenes, e.g., thymol and carvacrol, whose amount can be assumed as an indicator of the antimicrobial activity [14], and both open-chain and monocyclic monoterpenes with an allelopathic effect [15]. Notably, carvacrol has been approved for food use and classified as GRAS (generally recognized as safe) [16], while non-oxygenated monoterpenes, such as linalool and *p*-cymene, have shown analgesic effect [10]. A high content of the latter has been also associated with a significant role in plant protection from the herbivores [17,18]. The complementary fraction of non-volatile and more polar compounds includes variable amounts of secondary metabolites, such as phenolic acids [19], phenylpropanoids, fatty acids [20], tannins [3], and tocopherols [18]. A list of the main components of SEO is reported in Figure 1.

Nevertheless, a strong difference in the content of these components was reported, thus confirming the existence of several plant chemotypes [21]. The wide phytochemical diversity recorded in EOs is related to environmental factors, such as geographic and growth conditions, and to the developmental stages as isolation techniques and drying methods [22,23].

Recently, the emergence of antibiotic resistance mechanisms has highlighted the importance of the EOs as effective alternative to overcome this problem. Several studies reported that EOs, derived from different species of *Satureja* genus, possess remarkable antibacterial activities against different microorganisms such as *Escherichia coli*, *Listeria* spp., and *Staphylococcus* spp. [19,24,25]. This antimicrobial range and activity are likely dependent on the composition of the essential oil and the origin of the plant. Furthermore, a broad bactericidal activity of herbal plant oil nanoemulsions (NEs), impairing functional groups in the cell wall, lipids, proteins and nucleic acid towards both drug resistant Gram-positive and Gram-negative bacterial pathogens, has been described [26].

In this work, the chemical profile of four different *Satureja montana* L. EOs, three from growing wild populations (SEO1–3) and one from commercial source (SEOT), but different for ecological factors like the growing region and field altitude, has been investigated. The high sensitivity and unrivaled mass accuracy and resolving power of Fourier-transform ion cyclotron resonance mass spectrometry (FT-ICR MS) coupled with electrospray ionization have been exploited here to reveal (semi) polar compounds in an untargeted metabolomics approach, recently shown to cover a large number of molecular families in complex biological samples [27,28]. This approach allows reliable determination of molecular formulas thus offering a fast method to view the complex chemistry of biological matrices. However, the method is not apt to gain quantitative information due to potential ionization suppression and dissimilarities in detection sensitivity. Thus only an overall qualitative description of the metabolomics fingerprint can be gathered. In this perspective, the less polar volatile fraction has been characterized by gas chromatography (GC) MS, which allows to separate and identify structural isomers and to determine their relative abundances. Finally, in order to obtain a stable nanocarrier, different SEO NEs were prepared and characterized. Two different surfactants (Tween 20 and Tween 80) were chosen to stabilize NEs. Tweens are polysorbate-type nonionic surfactant formed by the ethoxylation of sorbitan before the addition of lauric acid (Tween 20) or oleic acid (Tween 80). Tween stability and relative nontoxicity allows them to be used in a number of cosmetic and pharmaceutical products. Tween 20 and Tween 80, bearing the same polar head but different apolar tail (Figure 2), show different HLB value. Different Tweens were chosen in order to evaluate how their features could affect NE physical–chemical characteristics. In particular, stability studies of NEs were carried out in order to evaluate their changes in size and ζ-potential at different temperature for a period of 90 days. Oil drop properties were investigated by fluorescence measurements employing Pyrene as fluorescent probe loaded inside NEs. Furthermore, size values obtained by DLS analyses were compared with data obtained by TEM observations. In order to determine the influence of chemical composition on bacterial growth, the minimum inhibitory concentration (MIC) and the minimum bactericidal concentration (MBC) of SEOs, alone or formulated in NEs, towards several clinical Gram-positive and Gram-negative strains, were evaluated. 

## 2. Materials and Methods

All chemicals were research grade products, supplied by a commercial source (Sigma Aldrich, Milan, Italy) and used without further purification.

### 2.1. Plant Material

The present study was performed on four different SEOs. SEOT was purchased from a commercial source (Talia, Roma, Italy, http://www.taliaessenze.com) that collected the plant material from Albania. SEO1, 2, and 3 were extracted in laboratory from the raw plants. Aerial parts of full blooming plants were harvested in October 2016 and the essential oil SEO1 was obtained in March 2017. Plants were collected at about 800–900 m above sea levels (a.s.l.) with SE exposure at two accesses in Abruzzo region (central Italy): Ernici and Gran Sasso Mountain. Sample identity was determined by L. Menghini. SEO2 and SEO3 oils were obtained from the plant (leaves and flowers) grown at 500–600 a.s.l. in Collepardo, Lazio region (central Italy) by the Sarandrea Marco and Co. s.r.l., Collepardo, (FR), Italy; http://www.sarandrea.it: SEO2 was prepared in December 2016 from the aerial parts of the plant harvested in October 2016, SEO3 was prepared in March 2018 from the aerial parts of the plant harvested in September 2017. The latter oil was available in larger amount for the experimental tests. The fresh material (leaves and flowers) was extracted by hydro-distillation for 180 min in a Clevenger-type apparatus and then decanted. The decanted oil was subsequently dried by addition of anhydrous sodium sulfate, stored in a dark glass bottle and stocked at −20 °C in order to prevent photo and thermo-degradation processes. The essential oil yield turned out to be 0.59 and 0.66% (*w*/*v*) for the Ernici and Gran Sasso samples, respectively.

### 2.2. Electrospray Ionization FT-ICR MS Analysis

A small amount (ca. 40 μg) of each SEO was filtered through a 0.45 μm hydrophobic polypropylene Acrodisc filter, diluted in 1.5 mL MeOH:AcN (1:1 *v*/*v*) and further diluted in MeOH to a final concentration of 1.7 μg L^−1^. SEOT composition has been reevaluated here for comparison purposes [23]. The experiments were run on a Bruker BioApex 4.7 T FT-ICR mass spectrometer equipped with an electrospray ionization (ESI) source and a cylindrical infinity cell. MS analyses were performed in either positive (ESI+) or negative (ESI−) polarity mode by direct infusion at a flow rate of 120 µL h^−1^. Metabolites were assayed in the *m*/*z* range between 50 and 1250 *m*/*z* with an average mass resolving power, *m*/Δ*m* 50%, of 60,000 at *m*/*z* 400. ESI FT-ICR mass spectra were collected accumulating ions in the hexapole for 0.2–0.4 s. 100–200 scans with an acquisition size of 1 M were coadded for each analysis. Raw data were recorded by using the Xmass Analysis software package and data treatment was performed by use of DataAnalysis 3.4 (Bruker Daltonics). In negative mode operation, 2% *v/v* NH_3_ was added to improve ionization. In both ionization modes, metabolites known to be present in the mixtures were exploited as internal mass calibrants, including: (i) carvacrol, revealed as deprotonated [M – H]^−^ ion at *m*/*z* 149.09718, (ii) linalool, detected as sodiated [M + Na^+^]^+^ ion *m*/*z* 177.12500; and (iii) limonene, revealed as protonated [M + H^+^]^+^ ion *m*/*z* 137.13248, so achieving a mass accuracy below 1 ppm. Additional confirmation was gathered by using a 5 μM standard solution of a standard mixture of carvacrol, linalool, and limonene in methanol. For each internally calibrated FT-ICR mass spectra, only peaks with a signal/noise ratio higher than 4 were considered. All mass measurements are based on the “monoisotopic” ion. The so obtained *m*/*z* values were extracted and submitted to MassTRIX [29] for metabolite assignment. In ESI+, protonated, sodiated, and potassiated adducts were considered, while in ESI−, deprotonated and chlorinated adducts were evaluated, with a maximum allowed deviation of ±3 ppm. Less polar compounds were assayed by an atmospheric pressure chemical ionization (APCI) source coupled to a 2000 Q TRAP (Applied Biosystem) commercial hybrid triple-quadrupole linear ion-trap mass spectrometer with a Q_1_q_2_Q_LIT_ configuration. Further validation of peak assignments was obtained by collision induced dissociation (CID) experiments to obtain structural information and by cross-reference with online libraries, including Metlin [30] (https://metlin.scripps.edu) and ChemCalc [31] (http://www.chemcalc.org). The large number of assigned molecular formulas (about 2000) generated from each EO was used to construct van Krevelen diagrams [32], where the molar ratio between the number of hydrogen and carbon atoms (H/C) is plotted versus the molar oxygen to carbon ratio (O/C). This visualization tool allows to cluster the identified compounds in the major classes of expressed metabolites. Differences between samples were also highlighted by Venn diagrams, while the elemental composition (CH, CHN, CHNO, CHNOP, CHO, CHOP) of the SEOs has been depicted by histograms.

### 2.3. Gas Chromatography/Mass Spectrometry Analysis

The volatile fraction of the SEO1–3 has been investigated through a GC-MS analysis. About 1 μL of the filtered oil has been manually injected in an Agilent Technologies 6850 gas chromatograph coupled with an Agilent Technologies 5975 mass spectrometer, equipped with the non-polar capillary column HP-5MS (30 m × 0.25 mm × 0.25 µm). The gas-chromatographic parameters were set as follows: inlet temperature, 250 °C; injection mode, split (split ratio 40/1); flow rate of the helium carrier gas (99.995% purity), 1.0 mL/min; oven temperature starting from 40 °C, after 5 min raised to 200 °C at 5 °C/min, and kept at this final temperature for 60 min. Mass spectrometry parameters were set as follows: EI energy, 70 eV; solvent delay, 6 min; source temperature, 230 °C; quadrupole temperature, 150 °C; mass scan *m*/*z* range, 50–350 u.

The identification of the eluted compounds was carried out by applying three criteria: EI spectrum, Kovats index (KI), and injection of standard compounds. More in detail, the EI spectra were compared with those collected in both commercial (FFNSC 3) and free databases (NIST 11, Flavor2) by accepting a matching quality never below 80%; KIs were measured using a mixture of *n*-alkanes (C7–C40) with the same chromatographic setup, and then compared with values reported in the FFNSC 3 and NIST 11 databases; finally the injection of several commercial standards confirmed the peak assignments. The quantitative analysis of the EO volatile compounds was performed by manually integrating the GC peaks without any further correction. The internal standard (IS) method has been exclusively applied to determine the actual mass percentage of carvacrol and thymol in the SEO3, by using caryophyllene oxide, as IS.

### 2.4. Nanoemulsion Preparation

All the delivery systems were prepared with SEO1, 2, 3, and T and, due to the highest number of terpenes and terpenoids and availability in greater quantities, SEO3 was chosen as an election sample. Four NEs (A1, A2 and B1, B2) were prepared using different amounts of Tween 20 (Tw20) or Tween 80 (Tw80) and SEO3 (Table 1) in 5 mL of Hepes buffer (10^−2^ M, pH 7.4). The selected formulations were characterized by different Oil/Surfactant ratios: 1:1 for A1 and B1 and 1:2 for A2 and B2.

A fast preparation method was used to obtain all samples [33]. Briefly, the mixture composed by SEO, Tw20 or Tw80 and Hepes buffer (10^−2^ M, pH 7.4) was vortexed for about 5 min allowing the micro-emulsion formation. Each emulsion with microscale droplets was sonicated for 20 min at 50 °C using a tapered microtip operating at 20 kHz at an amplitude of 18% (Vibracell-VCX 400, Sonics, Taunton, MA, USA) to obtain NEs. At this stage, all formulations can be sterilized by using cellulose filters (0.22 micrometers) in accordance with Ph. Eur.

### 2.5. Dynamic Light Scattering Measurements

Droplet size distribution and the ζ-potential of the NEs were measured at the temperature of 25 °C by dynamic light scattering (DLS), using a Zetasizer Nano ZS90 (Malvern Instruments Ltd., Worcestershire, UK), equipped with a 5 mW HeNe laser (wavelength *λ* = 632.8 nm) and a digital logarithmic correlator. The polydispersity index (PDI) value was also determined in order to evaluate homogeneity of the size distribution, in particular a PDI value lower than 0.3 indicates a monodisperse population.

### 2.6. Transmission Electron Microscopy (TEM)

A drop of diluted NEs samples was adsorbed for 1 min onto carbon-coated copper grids and negatively stained for 10 s with 2% filtered aqueous sodium phosphotungstate adjusted to pH 7.0. After the excess of staining solution removing at the border of the grid with filter paper, preparations were observed with a Philips 208S transmission electron microscope at 80 kV.

### 2.7. NEs Stability Evaluation

To assess colloidal stability at different temperatures, the selected NEs formulations were stored at 4 and 25 °C for a period of 90 days. Samples from each batch were withdrawn at definite time intervals (1, 30, 60, and 90 days) and the *ζ*-potential and the mean of hydrodynamic diameter were determined as previously described.

### 2.8. Fluorometric Measurements

By fluorescent techniques the oil drop features (microviscosity and polarity) were investigated. Pyrene loaded NEs were prepared by adding Pyrene (4 mM) to NEs components (same preparation method described above). The lateral distribution and the mobility of the probe inside the oil phase were studied by fluorescence measurements. Pyrene is a florescence probe and its monomer exhibited a spectrum characterized by five emission peaks (I1–I5) while its excimer shows only one peak (IE). Pyrene, at opportune concentration, can form intramolecular excimer based on the viscosity of the probe microenvironment [34]. The fluorescence signals emitted by Pyrene loaded NEs suspension were scanned (*λ* = 350–550 nm) using luminescence spectrometer (LS5013, PerkinElmer, Waltham, MA, USA) [35].

### 2.9. In Vitro Release Study

NEs were prepared with Nile Red (NR) a lipophilic probe at the concentration of 0.32 mg·mL^−1^. The lipophilic compounds, as well as NR, were dissolved into the oil and then the buffer solution was added. The NR release study is aimed to simulate the release of a lipophilic model drug entrapped in the dispersed oily phase from nanoemulsions. The experiments were carried out using dialysis tubes (Molecular Weight cut-off 8000 and 5.5 cm^2^ diffusing area) and following Nile Red increase in external medium (Hepes buffer 10^−2^ M, pH 7.4/ethanol 50/50 *v*/*v*) at definite time intervals, for 24 h at 37 °C. The release medium was gently magnetically stirred during the experiment. Sample volumes (1 mL) were withdrawn from the solution to perform UV analyses and then reinserted back in the external medium. Released NR was detected by means of a spectrophotometer (Lambda 25, PerkinElmer, Waltham, MA, USA ) and the analyses were carried out evaluating the NR absorbance at 559 nm. Aliquots were analyzed immediately after sampling. All release experiments were carried out in triplicate [36].

### 2.10. Bacterial Strains

Gram-positive Listeria monocytogenes, Staphylococcus aureus (SA1 and SA2), Staphylococcus hemolyticus, and Gram-negative Escherichia coli, Klebsiella pneumoniae, Pseudomonas aeruginosa and Serratia marcescens isolates were obtained from a collection of clinical strains of Department of Public Health and Infectious Diseases, Sapienza University of Rome. Bacterial strains were grown in brain heart infusion broth (BHI) (Oxoid) and maintained as stock cultures in 15% glycerol-BHI at −80 °C. Antimicrobial susceptibility tests were performed using the VITEK2 system instrument (bioMérieux). Results were analyzed by the VITEK2 Advanced Expert System (AES) software on the basis of EUCAST breakpoints (http://www.eucast.org/clinical_breakpoints/).

### 2.11. Minimum Inhibitory Concentration (MIC) and Minimum Bactericidal Concentration (MBC) Assays

The MIC determination of SEOs and NEs was performed by the microdilution method according to the CLSI method (https://clsi.org). Density values from oil were performed as described by Andrade et al., 2014 [37]. Exponentially growing bacterial cultures were diluted to cell density 0.5 McFarland and 10 μL of bacterial suspension was inoculated in wells of 200 μL, final volume, in Muller Hinton Broth (MHB) (Oxoid); after 18–24 h at 37 °C, the growth was evaluated. For oils, the concentrations ranged from 50 to 0.048 mg·mL^−1^ in MHB plus Tween 80 (0.02%) to increase oil solubility, and TTC (2,3,5-triphenyl-2H-tetrazolium chloride) (0.05%) to determine bacterial growth. For NE formulations (A1 and A2) the twofold dilution started from 18.4 mg·mL^−1^ whereas for the NE preparations (B1 and B2) from 19.6 mg·mL^−1^. All assays were performed in triplicate. The plates were incubated at 37 °C for 24 h. The MIC of the compounds was the lowest concentration that inhibited the visible growth of cells. To determine MBC, defined as the lowest concentration of antimicrobials that kills 99.9% or more of the initial inoculum, 10 µL aliquots from wells, where no visible growth was observed (MIC, 2× MIC and 4× MIC), were plated on Mueller–Hinton agar medium, then incubated at 37 °C for 24 h.

### 2.12. Determination of the Effect of SEOs and NEs on T24 Cell by MTT Assay

The T24 human bladder cancer cell line was obtained from (ATCC, Rockville, MA, USA). The cells were cultured in Minimum Essential Medium medium (HyClone; GE Healthcare, Logan, UT, USA) supplemented with 10% heat-inactivated fetal bovine serum (FBS; SAFC Biosciences Inc., Lenexa, KS, USA), 100 U/mL penicillin and 100 mg/L streptomycin. Proliferation was evaluated by 3-(4,5-dimethylthiazol-2-yl)-2,5-diphenyltetrazolium bromide (MTT) assay. Briefly, 96-well plates were seeded with 1 × 10^4^/well and different amounts of SEOs were added to the cells, either as NEs or as free oil. After 24-h exposure of the cells to the different preparations, 20 μL of a 5 mg·mL^−1^ MTT solution in PBS was added to each well, and the plates incubated for 3 h. After the formazan crystals were dissolved by the addition 100 μL of dimethyl sulphoxide, optical density (OD) at 570 nm was determined with a spectrophotometer/fluorimeter microplate reader (PerkinElmer, Hopkinton, MA, USA) and the values measured for treated cells, compared to those of untreated controls.

### 2.13. Statistical Analysis

Each experiment was performed in triplicate and all values are reported as mean ± standard deviation (SD). The data were statistically analyzed by one-way analysis of variance (ANOVA). A *p* value ≤0.05 was considered statistically significant.

## 3. Results and Discussion

### 3.1. MS Metabolite Profiling

Nowadays, ESI-MS is recognized as a method of choice in metabolomics, due to its high speed and sensitivity [38,39]. In this study direct infusion ESI coupled with high resolution mass spectrometry were aimed at increasing the metabolome coverage to detect also (moderately) polar compounds that are present in only minor amount. In comparison to mass spectrometry coupled to liquid chromatography (LC-MS) typically used to quantify a narrow set of target compounds, the non-targeted metabolite profiling applied here is a fast and reliable screening tool for a thorough metabolic fingerprinting and chemotaxonomic description, allowing for isomeric structures. Isomer discrimination remains an issue that is not solved by the assignment of a molecular formula and needs to be specifically coped with.

Herein, direct infusion ESI FT-ICR MS has been applied to characterize four different SEOs, avoiding time-consuming preliminary chromatographic separations [40,41], rather exploiting the unsurpassed mass accuracy and resolving power of FT-ICR. Furthermore, CID experiments have provided additional structural information, allowing to identify metabolites in the absence of commercial standards, as commonly performed in the analysis of complex mixtures [42].

The elemental formulas identified in all SEO samples are listed in Appendix A, where each item includes the experimental *m*/*z* ratio, the assigned molecular formula and the corresponding theoretical *m*/*z* ratio, the mass error and the putative compound annotations. In the investigated *m*/*z* range, the four sampled SEOs present a wide range of annotated peaks reported in Appendix A where SEO1 and SEO2 present the highest (# 2922) and lowest (# 702) number of ascertained ions in ESI(+), respectively, whereas the number of peaks in ESI(-) appears more balanced among the different SEOs. Examples of the SEOs mass spectra obtained in ESI(+) are reported in Figure 3 as enlargements in the low mass range (*m*/*z* 90–450). In agreement with previous evidence, SEOs present a significant variability in the chemical composition (Appendix A) [43]. On the whole, all samples contain metabolites characteristic of SEOs, including carvacrol/thymol, *p*-cymene, camphene/limonene/terpinene, camphor, eucalyptol/borneol/terpinen-4-ol, linalool-oxide, eugenol, and saturated free fatty acids [21,44].

Linalool oxide, identified as sodiated ion at *m*/*z* 193.11975, is a common, prominent peak of SEO1, 2 and 3, while barely detected in the commercial oil SEOT from a population grown in Albania already shown/found to be a linalool poor chemotype [21]. Although the pharmaceutical anxiolytic activity [45] is related to the non-oxygenated precursor linalool, linalool oxide has shown additional antinociceptive and anticonvulsant effects on animal models [10], pointing to a higher biological potential of SEO1, 2, and 3 with respect to SEOT. Conversely, SEOT is rich in choline (*m*/*z* 104.10701), an essential nutrient revealed also in SEO2 and SEO3. All SEOs present vitamin D derivatives, like methyl- and dihydroxy-vitamin D3, while SEO1 and SEOT hold the carotenoid zeta-carotene, and retinol (C_20_H_30_O) and retinal (C_20_H_28_O) vitamins, respectively. As reported in a preliminary contribution for SEOT [24], all the samples from wild growing populations SEO1–3 contain: (i) terpenes, like γ-terpinene revealed as [C_10_H_17_]^+^ in ESI(+) at *m*/*z* 137.13245, and related compounds, i.e., *p*-cymene, detected as [C_10_H_15_]^+^ in ESI(+) at *m*/*z* 135.11693; (ii) terpenoids, including carvacrol, and thymol, found as deprotonated species [C_10_H_14_O-H]^−^ at *m*/*z* 149.09717, as shown in the mass spectrum of SEO3 (Appendix A); (iii) free fatty acids (FAs), including saturated medium-long chain FAs, i.e., myristic (14:0), palmitic (16:0) and stearic (18:0) acids, at *m*/*z* 227.20182, 255.23239 and 283.26299, respectively, and monounsaturated FA, i.e., oleic (18:1) acid at *m*/*z* 281.24859. In addition, FA derivatives responsible for specific flavor development [46], including alcoholic (hexenol) and aldehydic (hexanal and octanal) products, were revealed either in SEO3 or in SEO1 and SEO2, respectively. Experiments carried out by APCI-QTRAP MS have allowed to magnify the ionization of less polar compounds and to conduct subsequent CID assays, as in the case of the peak at *m*/*z* 155 assigned to protonated terpenoids [C_10_H_19_O]^+^ (Appendix A), whose dissociation pattern is compatible with borneol and linalool, as previously reported for SEOT [24]. Other metabolites were also characterized by the same procedure, including [limonene + H]^+^ and [limonene aldehyde − H]^−^ as shown in Appendix A, respectively. To collect information from the significant number of assigned formulas, all the identified compounds were clustered in individual regions based on the H/C and O/C molecular ratio in van Krevelen diagrams illustrated in Figure 4. Overall, the most populated classes of metabolites, denoted by highlighted regions, correspond to: (i) lipids, including fatty acids and steroid-like compounds; (ii) polyphenols; and (iii) terpenoids. Not unexpectedly, carbohydrates are not largely represented due to the privileged extraction of volatile components from the physical chemical procedure applied for the preparation of the SEOs. Still, also less volatile compounds (e.g., fatty acids) contribute to the complex chemical composition of SEOs, as already documented [8]. It is obviously due to the high sensitivity of the FT-ICR mass spectrometric analysis that a remarkably rich array of compounds has been unveiled.

According to Figure 4, SEO1 displays the highest number of condensed hydrocarbons. The relative frequency distribution of the identified molecular formulas is reported in Figure 5 so as to obtain a qualitative evaluation of the elemental distribution in the sampled SEOs.

All SEOs contain a majority of CHO compounds, including FAs, alcohols, phenols, many polyketides, and terpenoids, which are responsible of pharmaceutical activity and flavor, followed by CHNO features, such as amino acids, amino-sugars, and amino FAs. SEOT and SEO3 present the highest number of CHO compounds (52%–53%) especially populated by terpenoids and lipids according to Figure 4, while SEO2 presents the smallest one (45%). SEO3 may be likely considered as the wild growing sample with the highest variability of terpenes and terpenoids. SEO2 presents the major biodiversity (30%) of CHNO components, while SEO3 presents the smallest one (23%). SEO1 contains the highest number of phosphorylated compounds (both CHOP and CHNOP). As a final remark, according to the notion that the terpenes are not easily ionized in ESI-MS, the amount of species in the CH class does not exceed 3% in all sampled SEOs, with small variations between SEO2 (1.5%) and SEOT (2.8%), as shown in Figure 4.

### 3.2. GC-MS Analysis of the Volatile Components

The GC-MS analysis of the SEOs (Appendix A) exhibited three substantially superimposable chromatograms (Table 2 and Appendix A). The detected phytochemicals have been classified as terpenes (24.7%–30.6%), terpenoids (66.5%–72.8%), and sesquiterpenes (0.4%–0.5%). In more detail, the compounds with the highest relative abundance are the SEOs are carvacrol (23.9%–29%), its isomer thymol (14.5%–16.5%), linalool (16.0%–16.4%), γ-terpinene (12.3%–15.0%), *p*-cymene (9.4%–12.3%), and eucalyptol (6.2%–6.8%). We observed a carvacrol/thymol (C/T) ratio [47] significantly different if compared to evidences previously published [43]. Furthermore, the direct comparison between the SEOs1–3 and the commercial SEOT [23] (Table 2) indicates several significant differences: (i) the carvacrol/thymol (C/T) ratio [47] of the SEOs1–3 ranges between 1.6–1.8, differently from the value of 5.8 measured in the SEOT. Nevertheless, the SEOT C/T ratio may be reduced to 3.7, if thymol and its methyl ether, absent in the SEOs1–3, are grouped in the relevant calculation. (ii) Linalool and eucalyptol, significantly abundant in the SEOs1–3, are totally absent in the SEOT. (iii) γ-terpinene relative abundance detected in the SEOs1–3 is approximately three times that observed in the SEOT sample. It is worth noting that, despite all the mentioned dissimilarities, the relative abundance of the three classes of compounds is very close to each other.

Finally, the absolute quantitative analysis of the carvacrol and thymol content in the SEO3 has been performed to evaluate their actual relevance in the overall composition of the sample. Interestingly, the real mass percentages (Table 3) are very close to the relative abundances resulting in the GC-MS reported in Table 2, indicating the crucial importance of the volatile fraction in the analyzed EO.

### 3.3. Characterization of Satureja Montana L. Essential Oil Nanoemulsions

All the delivery systems were prepared with SEO1, 2, 3, and T and, due to the highest number of terpenes and terpenoids and availability in greater quantities, SEO3 was chosen as an election sample among the wild growing species. Chemical physical characterization parameters are reported in Table 4 and Table 5. Chemically the DI-FTICR-MS has shown that this oil possesses the high variance of terpenes and terpenoids between the natural samples. DLS measurements (Figure 6) indicate that different surfactant ratios (Table 1) lead to NEs with different sizes (Table 4) but it can be pointed out that increasing surfactant content the dimension decrease. The dimensions of NEs in Hepes buffer were in the same range if the same oil/surfactant amount was employed (A1–B1; A2–B2). Electron microscopy observations (Figure 7) showed that all samples appeared as almost spherical in shape and with sizes corresponding to those revealed by DLS analysis. Furthermore, all samples show negative ζ-potential values.

NEs stability is a complex issue and involves chemical stability and physical stability which are all inter-related. The evaluation of parameters like stability (Figure 8 and Figure 9) or NEs oil phase characterization (Table 5) are fundamental to determine the potential future applications. Stability studies were performed according to the method previously described. In particular, A1 and B1 NEs are stable for at least 3 months when stored at both experimental temperatures. A2 and B2, characterized by a higher amount of surfactant, show a not significant dimension increase, more evident at 25 °C. All samples show negative ζ-potential values, increasing over time, able to prevent significant vesicle aggregation and promote NEs stability.

The classical DLVO theory has been developed for rigid colloids and no-DLVO forces should be taken into account in the study of very hydrophilic system, such as liposome or oil nanodroplet surfaces.

Amphipilic “shells” are able to be deformed and this feature should have an important effect on the van der Waals potential at the nanosystem surface [48,49].

Negative values of zeta potential of nanoemulsion stabilized with non-ionic surfactant could be associated with the selective adsorption of hydroxyl ion at the oil–water interface [50] as a result of the formation of hydrogen bond between oxyethylene group of Tweens and water molecules [51]. Furthermore, Roger et al. reported the presence of anionic impurities in emulsified systems causing the negative values of zeta potential [52].

The increase of zeta potential values over time could be related to modified adsorption of hydroxyl ions and/or anionic impurities at the oil/water droplet interface, resulting in a different interaction between water molecules and surfactant polar heads.

Oil phase studies were carried out by fluorescence analyses as previously described. The qualitative and quantitative effect of different surfactants employed for NEs on features of the oil phase, were investigated by Pyrene experiments and obtained data are shown in Table 5. Polarity values of NEs were similar for all samples, thus indicating that the presence of the two different surfactants (Tween 20 or Tween 80) has no effect on the oil phase microviscosity and polarity as well as their different concentrations. It is possible to conclude that stability, microviscosity and polarity trend are mainly related to the nature of the oil phase.

Release studies were also carried out as previously described. Release profiles of NR by NEs are shown in Figure 10. From the obtained results, it is possible to affirm that a certain amount of NR present in the formulation is released in the acceptor compartment. The entrapment efficiency of NR in the NEs is low and it affects the percentage of release. This behavior depends on chemical characteristics of the probe.

### 3.4. Antimicrobial Susceptibility Assays

Antimicrobial susceptibility tests were performed using the VITEK2 system. Antibiotic resistance profile is showed in Table 6. Most of the strains are multi-drug resistant (MDR) (3 ≥ classes). Staphylococcal strains are all resistant to penicillinase-stable penicillins (Oxacillin). *L. monocytogenes* was resistant to daptomycin, an antibiotic specific for Gram-positive infections.

### 3.5. Antibacterial Activity of EOs and NEs

The results confirm the SEOs activity against multidrug bacterial strains. According with other studies [24,53,54], as reported in Table 7, MIC and MBC values ranged from 0.39 to 6.25 mg·mL^−1^. Commercial SEOT showed the lowest MIC and MBC values towards Gram-positive and Gram-negative bacteria. Among the lab extracted, the SEO3 showed the stronger antimicrobial activity against bacterial strains. MIC and MBC values were often coincident. The results showed similar antibacterial activity of the oil toward *S. aureus*, both clinical and reference strains, with MIC values in the range 0.39–0.78 mg·mL^−1^ and a MBC of 0.78 mg·mL^−1^ (Table 7). Against *L. monocytogenes*, MIC was in the range 0.78–1.56 mg·mL^−1^; MBC was a concentration over the MIC value. Regarding *E. coli* strains, MIC and MBC values for oil coincided in the range 1.56–3.12 mg·mL^−1^. Antimicrobial activity of the members of *Satureja* genus is well known [3,24,25]. Significant antimicrobial activity of SEO has been previously related to its content in secondary metabolites, including carvacrol, thymol and terpinen-4-ol, probably through membrane damage [19]. Although the mechanism of antimicrobial activity of terpenes is not entirely known, it seems that these lipophilic compounds may alter the structural and functional integrity of the cell membrane in Gram-negative and Gram-positive bacteria as well as in fungi.

MIC and MBC for SEOs. SEOT: commercial oil; SEO1, SEO2 and SEO3_:_ lab extracted essential oils. Results are expressed in mg·mL^−1^.

NEs SEO3 formulation was able to preserve and improve antimicrobial activity compared to essential oil alone. As showed in Table 7 and Table 8, NE formulations induced a decrease of MIC for *S. aureus*, *E. coli*, and *S. marcescens*, suggesting a promising bactericidal activity towards some Gram-positive and Gram-negative bacterial pathogens [26].

### 3.6. Cytotoxicity Assays

Several studies described cytotoxic activity of essential oils and NE formulations towards tumor cell lines from different origins included human bladder cells [55,56].

Among possible SEO NEs application, use of this formulation as antiseptics, able to coat medical device as catheters in the treatment of catheter-associated urinary tract infections, could be useful.

Using MTT assay, cytotoxic activities of SEO3 oil and NEs were assessed on human bladder cells. Growth of cell line was inhibited in a dose-related manner after 24 of exposure to the essential oil. Values above 50 µg·mL^−1^ were considered significant cytotoxic concentrations (S45). Survival of cells exposed to SEO3 complexed into NEs was significantly higher than that detected in the samples treated with the same amount of free oil, indicating that the toxic effect of SEO was reduced in oil/surfactant NE preparation. In particular, at 50 µg·mL^−1^, compared to untreated control, cytotoxicity value decreased from about 50% to 25% when SEO3 was in NE (data not shown). Several studies have shown significant cytotoxic activity of carvacrol and thymol against different cell lines [57,58,59,60]; furthermore, some authors have shown that minor components play a role in different activity, possibly by producing synergistic effects, then contributing properties of the other constituents cannot be ruled out [61].

## 4. Conclusions

The high accuracy and sensitivity of the FT ICR-MS is confirmed to be a powerful tool for an extensive investigation of complex mixtures, like in the present study on SEOs. The high number of secondary metabolites, including terpenes and terpenoids, polyalcohols and fatty acids can be considered a key point to justify the SEO activity. At the same time, the notable variability in the composition of the (less) polar metabolites, as determined by direct infusion FT-ICR MS, as well as the comparatively higher abundance of carvacrol and thymol as revealed by GC-MS, leads us to ascribe most of the SEOs antimicrobial properties to these phenolic compounds. Furthermore, appropriate mixing of SEOs and surfactants enabled the formation of O/W nanoemulsions with droplet hydrodynamic diameter in the nanometric range, confirmed by DLS and TEM analyses and negative ζ-Potential able to assure high storage stability of the proposed nanoformulations.

Further studies are needed to recognize the contribution of selected molecular components of essential oils to the antimicrobial effect, to improve diffusion and persistence, and to characterize plant diversity not yet fully considered in order to explore the potential applications of this material for pharmaceutical purposes [62].

SEO NEs may represent a promising solution active against Gram-positive and Gram-negative clinical isolates. Further studies are in progress to load SEO NEs with active compounds to improve antibacterial activity. However, SEO NEs described in this study appear already suitable to be applied to surface disinfection practices in which could contribute to reduce environmental impact of drugs and antibiotic resistance. All in all, nanoemulsions have a number of potential benefits over other types of delivery systems. For example, they can be obtained from a wide range of food and pharmaceutical grade ingredients, such as oils, essential oils and emulsifiers, that are relatively inexpensive. Furthermore, NEs can be produced on a commercial scale using specific processing operations that are already commonly used in the pharmaceutical and food industry, such as high-pressure homogenization, microfluidization, and sonication. Obviously, scale-up is critical for developing commercially viable products, and hence, a nanoemulsion formulation should also be tested at the pilot-scale. The processing conditions for SEO NE production were optimized at the laboratory-scale by adjusting the formulation and processing time. A key challenge will be achieving similar hydrodynamic conditions to those of the laboratory-scale process to test NEs potential for pilot-scale production.

## Figures and Tables

**Figure 1 pharmaceutics-12-00007-f001:**
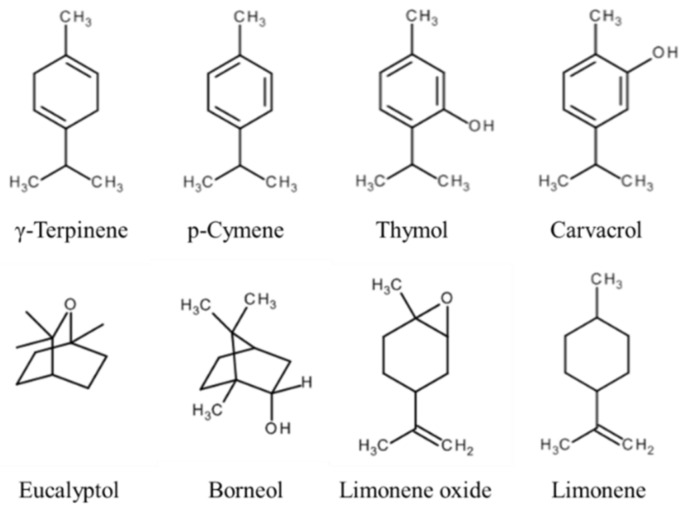
Most represented terpenes and terpenoids found in *Satureja montana* L. essential oil (SEO): Terpenes (γ-terpinene and *p*-cymene), oxygenated monoterpenes (thymol and carvacrol), bicyclic terpenoids (eucalyptol and borneol), and cyclic monoterpenes (limonene and limonene oxide).

**Figure 2 pharmaceutics-12-00007-f002:**
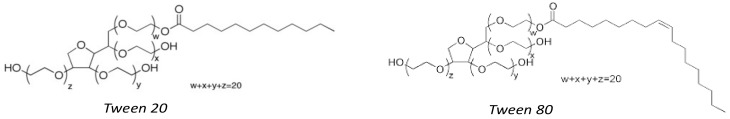
Tween 20 and Tween 80 surfactants employed to stabilize nanoemulsions (NEs).

**Figure 3 pharmaceutics-12-00007-f003:**
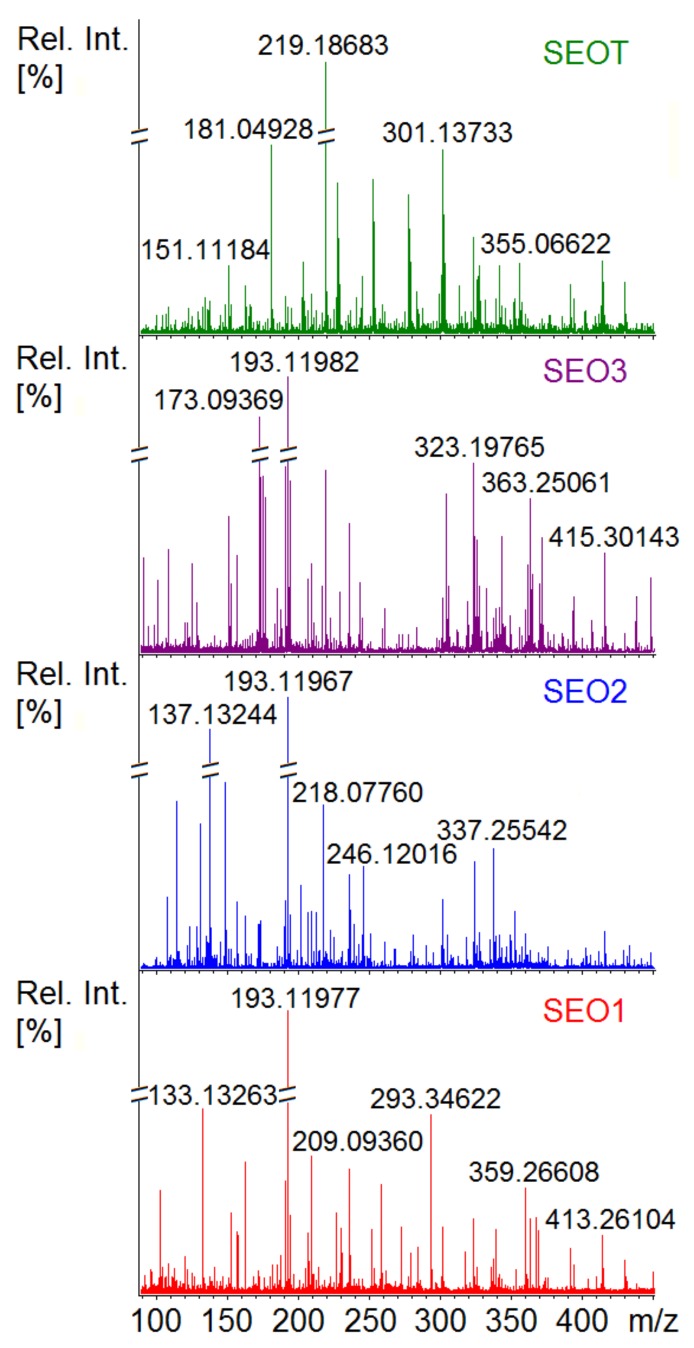
High resolution ESI(+) Fourier-transform ion cyclotron resonance (FT-ICR) mass spectra of the SEOs samples (red spectrum, SEO1; blue spectrum, SEO2; purple spectrum, SEO3; green spectrum, SEOT) are shown in the enlarged range of *m*/*z* 90–450.

**Figure 4 pharmaceutics-12-00007-f004:**
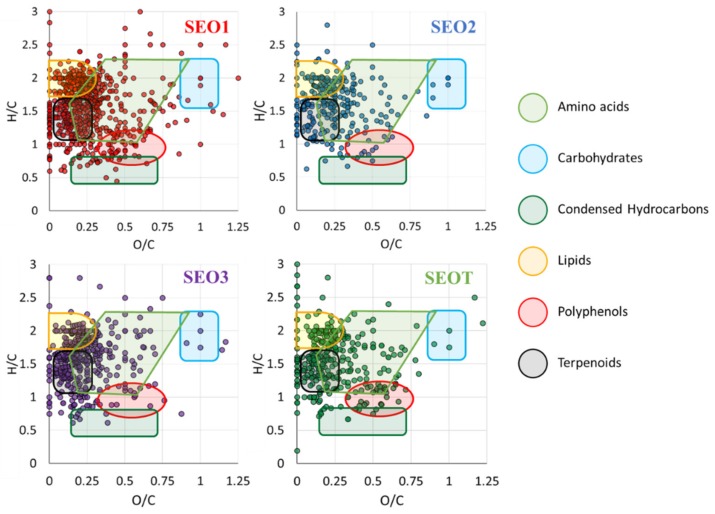
Van Krevelen diagrams of SEO1–3 and SEOT. The major metabolite classes involved in significant changes are lipids, terpenoids, and polyphenols. Terpenes (not highlighted in the legend) lie on the y axis.

**Figure 5 pharmaceutics-12-00007-f005:**
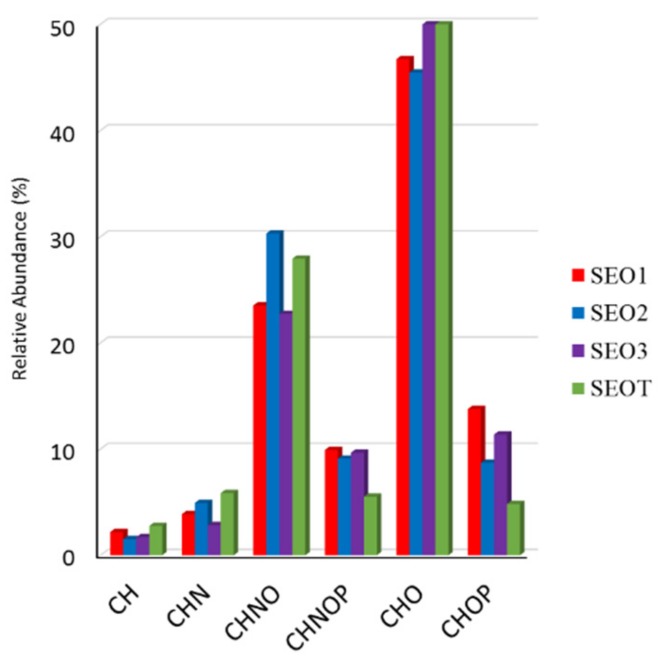
Elemental composition of the annotated metabolites in the SEOs samples.

**Figure 6 pharmaceutics-12-00007-f006:**
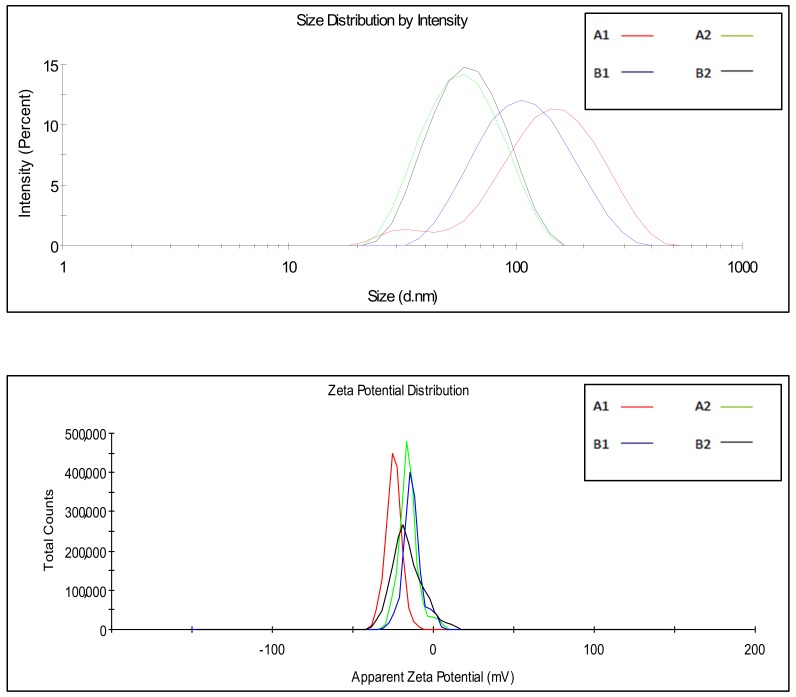
NEs DLS measurements.

**Figure 7 pharmaceutics-12-00007-f007:**
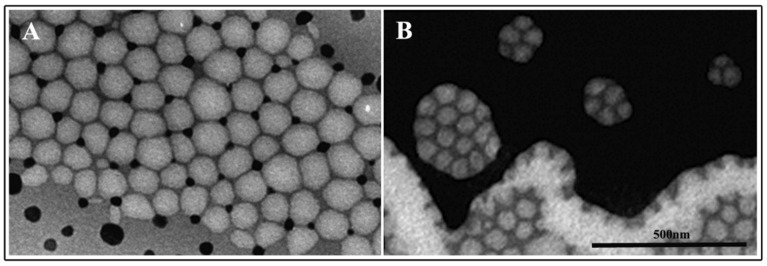
Transmission electron microscopy of SEO NEs. Panel (**A**) was representative of A1 and B1 nanoemulsions whereas panel (**B**) was representative of A2 and B2 NEs.

**Figure 8 pharmaceutics-12-00007-f008:**
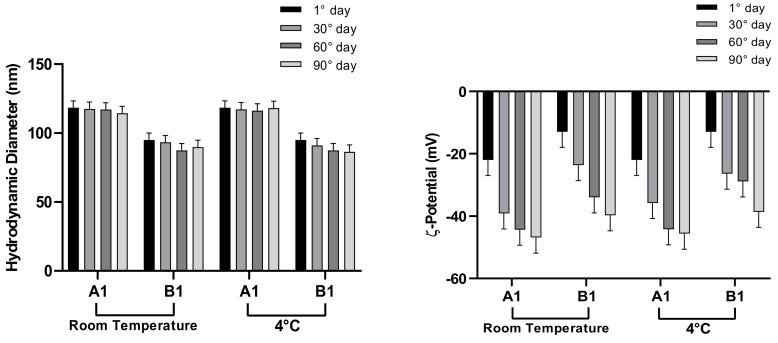
Stability studies of A1–B1 NEs at 25 and 4 °C.

**Figure 9 pharmaceutics-12-00007-f009:**
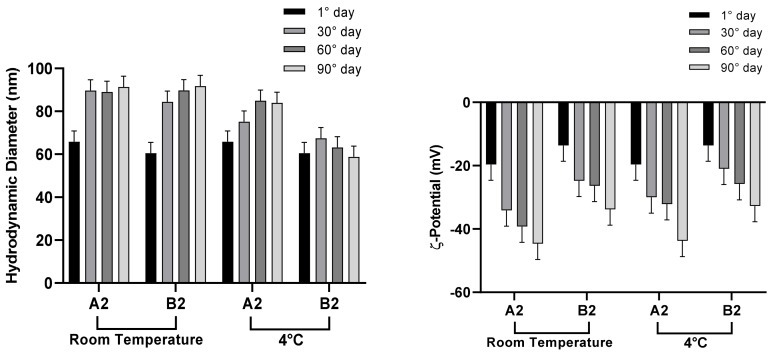
Stability studies of A2–B2 NEs at 25 and 4 °C.

**Figure 10 pharmaceutics-12-00007-f010:**
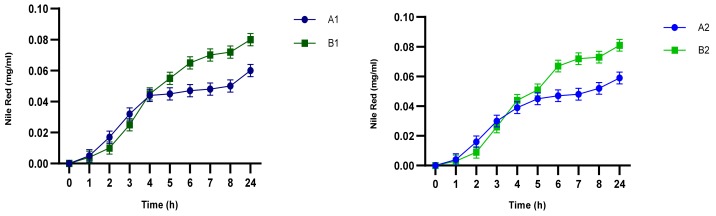
Nile red release profile by NEs.

**Table 1 pharmaceutics-12-00007-t001:** NEs compositions.

Samples	SEO3 [a]	Tw20 [a]	Tw80 [a]
A1	50	50	
A2	33.3	66.6	
B1	50		50
B2	33.3		66.6

[a] % *w*/*w*.

**Table 2 pharmaceutics-12-00007-t002:** Percentage of volatile components of SEO 1-3 and SEOT.

Class	Compound	SEO1	SEO2	SEO3	SEOT ^(a)^
Monoterpene	α-thujene	-	-	-	0.1
	α-pinene	0.4	0.4	0.4	1.3
	camphene	0.1	0.1	0.2	0.4
	β-pinene	0.2	0.2	0.2	-
	β-myrcene	0.1	0.1	0.5	0.6
	3-carene	-	-	-	0.2
	α-terpinene	-	-	-	0.8
	*p*-cymene	10.3	9.4	12.3	15.3
	limonene	1.3	1.3	1.8	0.9
Terpenoid	eucalyptol	6.2	6.3	6.8	-
Monoterpene	ocimene	-	-	0.2	-
	γ-terpinene	12.3	14.7	15.0	4.5
Terpenoid	linaloxide (cis)	0.4	0.2	0.2	-
Monoterpene	terpinolene	-	-	-	0.1
Terpenoid	linaloxide (trans)	0.3	0.2	0.1	-
	linalool	16.1	16.0	16.4	-
	camphor	0.2	0.2	0.3	0.1
	borneol	-	-	0.5	3.1
	terpinen-4-ol	-	-	-	1.2
	γ-terpineol	-	-	0.2	-
	α-terpineol	1.5	1.5	1.6	-
	thymolmethylether	-	-	-	4.3
	D-carvone	1.6	1.5	1.6	-
	thymol	16.5	16.1	14.5	7.6
	carvacrol	29.0	28.7	23.9	43.9
	eugenol	1.0	0.8	0.4	-
Sesquiterpene	β-caryophyllene	0.1	0.2	0.4	3.4
	α-humulene	-	-	0.1	-
	caryophylleneoxie	0.4	0.2	-	-
	unidentified	2.0	1.9	2.4	12.2
Monoterpene		24.7	26.2	30.6	24.2
Terpenoid		72.8	71.5	66.5	60.2
Sesquiterpene		0.5	0.4	0.5	3.4

^(a)^ See ref [23].

**Table 3 pharmaceutics-12-00007-t003:** Quantitative analysis of thymol and carvacrol in SEO3.

Compound	Mass Percentage
carvacrol	29.9
thymol	16.0

**Table 4 pharmaceutics-12-00007-t004:** Hydrodynamic diameter (HD), ζ-potential and polydispersity index (PDI) values of NEs.

Samples	HD (nm) ± SD	ζ-Pot (mV) ± SD	PDI ± SD
A1	118.40 ± 0.61	−19.00 ± 0.21	0.26 ± 0.01
A2	55.86 ± 0.93	−16.61 ± 1.06	0.18 ± 0.03
B1	95.05 ± 1.55	−13.02 ± 0.36	0.23 ± 0.01
B2	56.08 ± 1.17	−14.14 ± 0.25	0.17 ± 0.01

**Table 5 pharmaceutics-12-00007-t005:** NE oil phase characterization.

Sample	I_E_/I_3_ (Microviscosity) ± SD	I_1_/I_3_ (Polarity) ± SD
A1	4.49 ± 0.07	1.62 ± 0.13
A2	4.04 ± 0.16	1.47 ± 0.23
B1	4.69 ± 0.11	1.44 ± 0.07
B2	3.79 ± 0.21	1.50 ± 0.06

**Table 6 pharmaceutics-12-00007-t006:** Antibiotic resistance profile.

Bacterial Strain	Antibiotic Resistance Profile
*L. monocytogenes*	Oxacillin, Levofloxacin, Daptomycin
*S. aureus* (SA1)	Erythromycin, Fosfomycin, Gentamicin, Levofloxacin, Oxacillin, Rifampicin, Tetracycline, Sulfamethoxazole/Trimethoprim
*S. aureus* (SA2)	Benzylpenicillin, Clindamycin, Erythromycin, Levofloxacin, Oxacillin
*S. haemolyticus*	Clindamycin, Erythromycin, Fosfomycin, Gentamicin, Levofloxacin, Oxacillin, Sulfamethoxazole/Trimethoprim
*E. coli*	Full sensitive
*K. pneumoniae*	Amoxicillin/Clavulanate, Cefepime, Cefotaxime, Ceftazidime, Ciprofloxacin, Fosfomycin, Gentamicin, Sulfamethoxazole/Trimethoprim
*P. aeruginosa*	Amoxicillin/Clavulanate, Cefotaxime, Ertapenem, Fosfomycin, Tigecycline, Sulfamethoxazole/Trimethoprim
*S. marcescens*	Amoxicillin/Clavulanate, Colistin

**Table 7 pharmaceutics-12-00007-t007:** Minimum inhibitory concentration (MIC) and the minimum bactericidal concentration (MBC) of different SEOs.

	SEO1		SEO2		SEO3		SEOT	
*Bacterial Strain*	MIC	MBC	MIC	MBC	MIC	MBC	MIC	MBC
*L. monocytogenes*	1.56	3.12	1.56	3.12	0.78	1.56	0.78	0.78
*S. aureus* (SA1)	3.12	3.12	1.56	3.12	1.56	1.56	0.78	0.78
*S. aureus* (SA2)	3.12	3.12	1.56	1.56	1.56	1.56	0.78	0.78
*S. haemolyticus*	3.12	3.12	1.56	1.56	1.56	1.56	0.78	0.78
*E. coli*	3.12	6.25	1.56	3.12	3.12	3.12	1.56	1.56
*K. pneumoniae*	1.56	3.12	1.56	3.12	0.78	0.78	0.39	0.39
*P. aeruginosa*	3.12	3.12	3.12	3.12	3.12	3.12	1.56	1.56
*S. marcescens*	3.12	3.12	1.56	1.56	0.78	0.78	0.39	0.39

**Table 8 pharmaceutics-12-00007-t008:** Antimicrobial activity of NEs against Gram-positive and Gram-negative bacterial strains.

*Bacterial Strain*	NE-A1		NE-A2		NE-B1		NE-B2	
	MIC	MBC	MIC	MBC	MIC	MBC	MIC	MBC
*L. monocytogenes*	0.78	1.56	1.56	1.56	1.56	1.56	1.56	1.56
*S. aureus* (SA1)	0.78	1.56	0.78	1.56	0.78	1.56	1.56	1.56
*S. aureus* (SA2)	0.78	1.56	0.78	1.56	0.78	1.56	1.56	1.56
*S. haemolyticus*	1.56	2.3	9.2	9.2	2.3	4.6	9.2	9.2
*E. coli*	0.39	0.78	1.56	1.56	0.39	1.56	1.56	1.56
*K. pneumoniae*	0.78	0.78	1.56	1.56	0.78	0.78	1.56	1.56
*P. aeruginosa*	4.6	4.6	9.2	9.2	9.2	9.2	9.2	9.2
*S. marcescens*	0.78	0.78	0.78	1.56	0.39	0.78	1.56	1.56

MIC and MBC for NEs (A1, A2, B1, B2) containing SEO3. Results are expressed in mg·mL^−1^.

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
