# Peer review of "Satureja montana L. Essential Oils: Chemical Profiles/Phytochemical Screening, Antimicrobial Activity and O/W NanoEmulsion Formulations"

_pharmaceutics, 2019, doi:10.3390/pharmaceutics12010007_

Round 1

Reviewer 1 Report

The reviewer was happy to see extensive improvement in the revised manuscript. My previous comments were addressed properly. Hence, the reviewer suggests acceptance of the manuscript. 

One concern is that regarding to the cytotoxicity-- The MIC and MBC values ranged from 0.39 to 6.25 mg/mL, however, the author presented that the samples are cytotoxic above 50 µg/ml which equals 0.05 mg/mL. Dose this mean that the current antibacterial activity are false positive results?

Author Response

The reviewer was happy to see extensive improvement in the revised manuscript. My previous comments were addressed properly. Hence, the reviewer suggests acceptance of the manuscript.

One concern is that regarding to the cytotoxicity-- The MIC and MBC values ranged from 0.39 to 6.25 mg/mL, however, the author presented that the samples are cytotoxic above 50 µg/ml which equals 0.05 mg/mL. Dose this mean that the current antibacterial activity are false positive results?

In our study, anti-bacterial activity of the essential oil was assessed using a microdilution method; an MTT cytotoxicity assay was employed to test effects of the oil on cell line. As demonstrated by some authors, a low dose of essential oil could be able to induce cytoxicity against the human cell line.

Reviewer 2 Report

The authors addressed all the evidenced issues and corrected the manuscript accordingly.

Author Response

The authors addressed all the evidenced issues and corrected the manuscript accordingly.

Reviewer 3 Report

In this article, the authors proposed the formulation of Satureja montana L. essential oils as O/W nanoemulsions. Obtained products were tested to evaluate their antimicrobial activity.

The paper could be of interest for the readers of Pharmaceutics. I suggest some modifications and integrations.

Comments:

Table 1: the NEs composition reported in table 1 represents the theoretical composition? Do you evaluate the effective composition? Which type of test do you used? Paragraph 2.9: the preparation of NEs with Nile Red is not clear. When you add the probe? Which is the external medium used for the release study? Paragraph 2.12: why do you chose the T24 cell line as cellular model? It is not clear. Discussion of obtained results is lacking. Why do you observed the mean diameter reduction of NEs after the addiction of higher amounts of surfactants? Why do you observed an increase of mean diameter after the preservation of A2 and B2 formulations? Which type of stabilizer can be used? It is possible to produce a stable product with the proposed technique? Overall, I think that it is very important to discuss the obtained results. In conclusion section author have speculated the possibility to use SEO nanoemulsions in food industry and in clinical studies? Do you evaluate the possibility to sterilize the product? Do you evaluate the possibility to produce on large scale the NEs? Which are the costs of these products? The conclusions need to be supported by the results or by other existed evidences. I suggest to implement the conclusion section with these informations.

Author Response

In this article, the authors proposed the formulation of Satureja montana L. essential oils as O/W nanoemulsions. Obtained products were tested to evaluate their antimicrobial activity.

The paper could be of interest for the readers of Pharmaceutics. I suggest some modifications and integrations.

Comments

Table 1: the NEs composition reported in table 1 represents the theoretical composition? Do you evaluate the effective composition? Which type of test do you used?

Authors thank the reviewer for the interesting considerations about the manuscript. Table 1 reports the theoretical and effective compositions of the samples because after equilibrium, the samples were visually assessed for clarity or turbidity, so it can be concluded that all the oil is dispersed in nanoemulsions even because no purification step was carried out.

Paragraph 2.9: the preparation of NEs with Nile Red is not clear. When you add the probe? Which is the external medium used for the release study?

The paragraph 2.9 In vitro release study has been modified according to reviewer suggestions.

Paragraph 2.12: why do you chose the T24 cell line as cellular model? It is not clear. Discussion of obtained results is lacking.

As required by the Reviewer the sentence has been added in paragraph 3.6:

Several studies described cytotoxic activity of essential oils and NE formulations towards tumor cell lines from different origins included human bladder cells (Iman Sadeghi et al. “In vitro Cytotoxic and Antimicrobial Activity of Essential Oil From Satureja Intermedia” Iran Red Crescent Med J. 2013 15(1): 70–74; Madson R. F. et al. “Citotoxic activity evaluation of essential oils and nanoemulsions of Drimys angustifolia and D. brasiliensis on human glioblastoma (U-138 MG) and human bladder carcinoma (T24) cell lines in vitro.” Rev. bras. farmacogn. 2013 23-2 Curitib).Among possible SEO NEs application, use of this formulation as antiseptics, able to coat medical device as catheters in the treatment of catheter-associated urinary tract infections, could be useful.

Why do you observed the mean diameter reduction of NEs after the addiction of higher amounts of surfactants? Why do you observed an increase of mean diameter after the preservation of A2 and B2 formulations? Which type of stabilizer can be used? It is possible to produce a stable product with the proposed technique? Overall, I think that it is very important to discuss the obtained results.

The two selected formulations are the result of a deep formulative study. It has been noticed that a higher amount of surfactant is able to reduce NE  droplets (Colloids and Surfaces A 557 (2018) 76–84), but at the same time this reduction causes a slight decreased stability over time probably due to a higher reactivity because of the increased  oil/water interface.

The dimension variations are not significant, so the samples can be considered stable.

In conclusion section author have speculated the possibility to use SEO nanoemulsions in food industry and in clinical studies?

Conclusion section has been enriched with considerations about commercial use of nanoemulsions.

Do you evaluate the possibility to sterilize the product?

NEs can be sterilized and in some experiment we already performed such sterilization, but in our specific case it is not required. The following sentence has been added to section 2.4:

“At this stage, all formulations can be sterilized by using cellulose filters (0.22 micrometers) in accordance with Ph. Eur ”.

Do you evaluate the possibility to produce on large scale the NEs? Which are the costs of these products? The conclusions need to be supported by the results or by other existed evidences. I suggest to implement the conclusion section with these informations.

Conclusion section has been implemented according to reviewer suggestions:

SEO NEs may represent a promising solution active against Gram-positive and Gram-negative clinical isolates. Further studies are in progress to load SEO NEs with active compounds to improve antibacterial activity. However, SEO NEs described in this study appear already suitable to be applied to surface disinfection practices in which could contribute to reduce environmental impact of drugs and antibiotic resistance. All in all, nanoemulsions have a number of potential benefits over other types of delivery systems. For example, they can be obtained from a wide range of food and pharmaceutical grade ingredients, such as oils, essential oils and emulsifiers, that are relatively inexpensive. Furthermore, NEs can be produced on a commercial scale using specific processing operations that are already commonly used in the pharmaceutical and food industry, such as high-pressure homogenization, microfluidization, and sonication. Obviously, scale-up is critical for developing commercially viable products, and hence, a nanoemulsion formulation should also be tested at the pilot-scale. The processing conditions for SEO NE production were optimized at the laboratory-scale by adjusting the formulation and processing time. A key challenge will be achieving similar hydrodynamic conditions to those of the laboratory-scale process to test NEs potential for pilot-scale production.

Reviewer 4 Report

The authors studied the fingerprints of four different Satureja montana L. essential oils (SEOs). FT-ICR MS and GC-MS were adopted to determine the polar metabolite composition and non-polar volatile fraction of the SEOs, respectively. The SEOs were further prepared into the nanoemulsion (NEs) to improve the solubility and delivery. The antimicrobial effect of SEOs and their NEs were confirmed with multi-drug resistant bacteria strains. The NEs exibited improved efficacy against S. aureus, E. coli and S. marcescens. This work provides valuable insight on the chemical composition of SEOs and underlines the possible variation among different plant resource. The following formulation screening and testing shows the application potential in food industry or animal farm. Despite the aforementioned importance, there are still a few questions need to be addressed before it can be considered for publication.

Comments:

In the in vitro release study (Figure 10), the authors used Nile red as the model molecule to simulate the release profile of SEOs from NEs. However, Nile red is intrinsically different with SEOs in terms of hydrophobicity and composition. From previous data, we know the compounds in SEOs is highly heterogeneous. Please provide further explanation. The authors tested the stability of the NEs by monitoring the size and zeta-potential as long as 90 days. In figure 8, although the size of the NEs kept constant, the zeta-potential of NEs became more negative over time, suggesting the changes in surface chemistry of the NEs. The authors might want to discuss more about the reason behind it. In table 5, the authors claim the NEs are stable due to the indifference of micro-viscosity and polarity. But only one measurement is performed, making it less convincing. Please multiply the measurements and perform the statistical analysis. By comparing Table 7 and Table 8, the NEs only shows improved efficacy against S. aureus, E.coli and S. marcescens. But not applicable for other stains and formulations. Thus it’s not appropriate to imply that the NEs SEO3 formulation was able to preserve and improve antimicrobial activity compared to essential oil alone. The label and figure caption in the following figures are not clear: figure 3 and figure 4.

Author Response

The authors studied the fingerprints of four different Satureja montana L. essential oils (SEOs). FT-ICR MS and GC-MS were adopted to determine the polar metabolite composition and non-polar volatile fraction of the SEOs, respectively. The SEOs were further prepared into the nanoemulsion (NEs) to improve the solubility and delivery. The antimicrobial effect of SEOs and their NEs were confirmed with multi-drug resistant bacteria strains. The NEs exibited improved efficacy against S. aureus, E. coli and S. marcescens. This work provides valuable insight on the chemical composition of SEOs and underlines the possible variation among different plant resource. The following formulation screening and testing shows the application potential in food industry or animal farm. Despite the aforementioned importance, there are still a few questions need to be addressed before it can be considered for publication.

Comments:

In the in vitro release study (Figure 10), the authors used Nile red as the model molecule to simulate the release profile of SEOs from NEs. However, Nile red is intrinsically different with SEOs in terms of hydrophobicity and composition. From previous data, we know the compounds in SEOs is highly heterogeneous. Please provide further explanation.

The paragraph 2.9 In vitro release study has been modified providing further sperimental details. In particular Nile Red, a lipophilic probe, has been used to mimic the release of a lipophilic model drug. Nile Red was dissolvend in SEO phase of nanoemulsions and the release over time was also evaluated.

The authors tested the stability of the NEs by monitoring the size and zeta-potential as long as 90 days. In figure 8, although the size of the NEs kept constant, the zeta-potential of NEs became more negative over time, suggesting the changes in surface chemistry of the NEs. The authors might want to discuss more about the reason behind it.

Discussion has been increased adding the following explanation:

The classical DLVO theory has been developed for rigid colloids and no-DLVO forces should be taken into account in the study of very hydrophilic system, such as liposome or oil nanodroplet surfaces.

Amphipilic “shells” are able to be deformed and this feature should have an important effect on the van der Waals potential at the nanosystem surface. (V. Labhasetwar, M. S. Mohan & A. K. Dorle (1994) A study on zeta potential and dielectric constant of liposomes, Journal of Microencapsulation, 11:6, 663-668, DOI: 10.3109/02652049409051117

Stella K. Tsermentseli, Konstantinos N. Kontogiannopoulos , Vassilios P. Papageorgiou and Andreana N. Assimopoulou Comparative Study of PEGylated and Conventional Liposomes as Carriers for Shikonin Fluids 2018, 3, 36; doi:10.3390/fluids3020036)

Negative values of zeta potential of nanoemulsion stabilized with nonionic surfactant could be associated with the selective adsorption of hydroxyl ion at the oilwater interface (Stachurski, J., & MichaŁek, M. (1996). The Effect of the ζ potential on the stability of a nonpolar oilinwater emulsion. Journal of Colloid and Interface Science, 184(2), 433436.) as a result of the formation of hydrogen bond between oxyethylene group of Tweens and water molecules (Liu, W., Sun, D., Li, C., Liu, Q., & Xu, J. (2006). Formation and stability of paraffin oilinwater nanoemulsions prepared by the emulsion inversion point method. Journal of Colloid and Interface Science, 303(2), 557563.). Furthermore, Roger et al. reported the presence of anionic impurities in emulsified systems causing the negative values of zeta potential (Roger, K., & Cabane, B. (2012). Why are hydrophobic/water interfaces negatively charged? Angewandte Chemie International Edition, 51(23), 5625–5628).

The increase of zeta potential values over time could be related to modified adsorption of hydroxyl ions and/or anionic impurities at the oil/water droplet interface, resulting in a different interaction between water molecules and surfactant polar heads.

In table 5, the authors claim the NEs are stable due to the indifference of micro-viscosity and polarity. But only one measurement is performed, making it less convincing. Please multiply the measurements and perform the statistical analysis.

As reported in section 2.13 each experiment was performed in triplicate, but the standard deviations are missing in table 5. Authors apologize for forgetting these informations that now have been added in table 5.

By comparing Table 7 and Table 8, the NEs only shows improved efficacy against S. aureus, E.coli and S. marcescens. But not applicable for other stains and formulations. Thus it’s not appropriate to imply that the NEs SEO3 formulation was able to preserve and improve antimicrobial activity compared to essential oil alone.

As correctly required by the Reviewer the sentence has been modified in: “As showed in Table 7 and 8, NE formulations induced a decrease of MIC for S. aureus, E.coli and S. marcescens, suggesting a promising bactericidal activity towards some Gram-positive and Gram-negative bacterial pathogens”.

The label and figure caption in the following figures are not clear: figure 3 and figure 4.

Following the Reviewer 4's suggestion, the labels and the captions of Figure 3 and 4 have been accordingly modified for the sake of clarity.

Reviewer 5 Report

Review of the manuscript: " Satureja montana L. essential oils: chemical profiles/phytochemical screening, antimicrobial activity and O/W NanoEmulsion formulations”

Manuscript ID - pharmaceutics-623925

Medicinal plants are promising and still not really well investigated source of therapeutic agents – against different illness. In my opinion the manuscript is really well presented. I do not have any doubts that it can be presented in Pharmaceutics. Below I have presented only several (minor of importance) suggestions and remarks.

Abstract – well written. I have only two remarks. In my opinion Gram-positive and Gram-negative bacteria should be listed separately (not only specialists in microbiology will read this manuscript) – lines 42-44. Moreover, I would suggest presenting some, most important results, especially MIC values for the most active products and formulations.  

Introduction 

Excellent part of the manuscript. The authors presented most important information about botanical source of investigated essential oils (Satureja montana), analytical techniques they used for investigation and two surfactants (Tween20 and Tween80) – used for new formulations of EO.

One small comment - Line 86 – there is information: “A summary of the main components of SEO is reported in Figure 1”. In my opinion only in this figure only structures of most important components are presented, thus “summary” is not a good word.

Materials and methods

The authors investigated one commercial EO and prepared three other products in laboratory. In my opinion it is important that plants were collected in different places. A standard procedure was used for preparing the oils from plant material (hydro-distillation). Important advantage of the manuscript is the fact that modern analytical methods were used for determination of chemical composition of the oils. Most important physicochemical parameters of the oils/nanoemulsion formulations were also determined with modern techniques. Biological activity of the products was investigated according to CLSI method and with MTT assay. All experiments were well planned and performed, I do not have any critical remarks to this part of the manuscript. I have only one question. The MIC values is presented as mg/ml, some authors use a bit different approach and MIC value is presented as % (volume/volume). Does it mean that you determined the density of the oils (mg/ml) – in my opinion it was necessary for determination of MIC values in mg/ml.

Results

The obtained results are very interesting and well presented – I have only one suggestion, the Table 6 should be moved to Material and methods.

Final decision – formally I proposed minor revision (the authors should consider my suggestions), but it is really an excellent manuscript.

Author Response

Medicinal plants are promising and still not really well investigated source of therapeutic agents – against different illness. In my opinion the manuscript is really well presented. I do not have any doubts that it can be presented in Pharmaceutics. Below I have presented only several (minor of importance) suggestions and remarks.

Abstract – well written. I have only two remarks. In my opinion Gram-positive and Gram-negative bacteria should be listed separately (not only specialists in microbiology will read this manuscript) – lines 42-44. Moreover, I would suggest presenting some, most important results, especially MIC values for the most active products and formulations.

In the Abstract Section Gram positive and Gram-negative bacteria have been listed separately in parenthesis:

Gram-positive (Listeria monocytogenes, Staphylococcus aureus, Staphylococcus haemolyticus) and Gram-negative clinical isolates (Escherichia coli, Klebsiella pneumoniae, Pseudomonas aeruginosa and Serratia marcescens).

As suggested by Reviewer the sentence “MIC and MBC values ranged from 0.39 to 6.25 mg.mL−1” was added in the Abstract Section.

Introduction 

Excellent part of the manuscript. The authors presented most important information about botanical source of investigated essential oils (Satureja montana), analytical techniques they used for investigation and two surfactants (Tween20 and Tween80) – used for new formulations of EO.

One small comment - Line 86 – there is information: “A summary of the main components of SEO is reported in Figure 1”. In my opinion only in this figure only structures of most important components are presented, thus “summary” is not a good word.

“Summary” has been changed with the word “list”.

Materials and methods

The authors investigated one commercial EO and prepared three other products in laboratory. In my opinion it is important that plants were collected in different places. A standard procedure was used for preparing the oils from plant material (hydro-distillation). Important advantage of the manuscript is the fact that modern analytical methods were used for determination of chemical composition of the oils. Most important physicochemical parameters of the oils/nanoemulsion formulations were also determined with modern techniques. Biological activity of the products was investigated according to CLSI method and with MTT assay. All experiments were well planned and performed, I do not have any critical remarks to this part of the manuscript. I have only one question. The MIC values is presented as mg/ml, some authors use a bit different approach and MIC value is presented as % (volume/volume). Does it mean that you determined the density of the oils (mg/ml) – in my opinion it was necessary for determination of MIC values in mg/ml.

As correctly indicate by the Reviewer, different approaches are used to determine and to present the MIC values. In “Material and Methods” has been added:

“Density values from oil were performed as described by Andrade et al., 2014”.

Results

The obtained results are very interesting and well presented – I have only one suggestion, the Table 6 should be moved to Material and methods.

As required by previous Reviewer during our first submission to Pharmaceutics journal, we moved Table 6 from “Material and Methods” to “Results” Section.

Final decision – formally I proposed minor revision (the authors should consider my suggestions), but it is really an excellent manuscript.

Round 2

Reviewer 3 Report

Authors significantly improved the manuscript.

Author Response

Authors thank the reviewer for the positive comment.